

# Application of machine learning to proximal gamma-ray and magnetic susceptibility surveys in the Maritime Antarctic: assessing the influence of periglacial processes and landforms

Danilo C. de Mello[1]; Clara G. O. Baldi[1]; Cássio M. Moquedace[1]; Isabelle de A. Oliveira[1]; Gustavo V. Veloso[1]; Lucas C. Gomes[3]; Márcio R. Francelino[1]; Carlos E. G. R. Schaefer[1]; Elpídio I. Fernandes-Filho[1]; Edgar B. de Medeiros Júnior[1]; Fabio S. de Oliveira[4]; José J. L. L. de Souza[1]; Tiago O. Ferreira [2]; José A. M. Demattê[2.]

[1] Department of Soil Science, Federal University of Viçosa: daniloc.demello@gmail.com; clara.baldi@ufv.br; cassio.moquedace@ufv.br; isabelle.angeli@ufv.br; gustavo.v.veloso@gmail.com; marcio.francelino@ufv.br; carlos.schaefer@ufv.br; elpidio@ufv.br; edgar.junior@ufv.br; jjlelis@ufv.br
[2] Department of Soil Science, "Luiz de Queiroz" College of Agriculture, University of São Paulo, Av. Pádua Dias, 11, CP 9, Piracicaba, SP 13418-900, Brazil; e-mails; toferreira@usp.br; jamdemat@usp.br
[3] Department of Agroecology, Aarhus University, Blichers Allé 20, 8830 Tjele, Denmark: lucas.gomes@agro.au.dk
[4] Department of Geography, Federal University of Minas Gerais: fabiosolos@gmail.com

*Correspondence to*: Danilo César de Mello (daniloc.demello@gmail.com)

**Abstract.** Maritime Antarctica (M.A.) contains the most extensive and diverse lithological exposure compared to the entire continent. This lithological substrate reveals a rich history encompassing lithological, pedogeomorphological, and glaciological aspects of M.A., all influenced by periglacial processes. Although geophysical surveys can detect and provide valuable information to understand Antarctic lithologies and their history, such surveys are scarce on this continent and, in practice, almost non-existent. In this sense, we conducted a pioneering and comprehensive gamma-spectrometric (natural radioactivity) and magnetic susceptibility (κ) survey on various igneous rocks. The main objective was to create ternary gamma-ray and κ maps using machine learning algorithms, terrain attributes, and a nested-leave-one-out cross-validation method. Additionally, we investigated the relationship between the distribution of natural radioactivity and κ to gain insights into pedogeomorphological and periglacial processes and dynamics. For that, we used proximal gamma-spectrometric and κ data in different lithological substrates associated to terrain attributes. The geophysical variables were collected in the field from various lithological substrates, by use field portable equipment. The geophysical variables were collected in the field from various lithological substrates using portable equipment. These variables, combined with relief data and lithology, served as input data for modeling to predict and spatially map the content of radionuclides and κ by random forest algorithm (RF). In addition, we use nested-LOOCV as a form of external validation in a geophysical data with a small number of samples, and the error maps as evaluation of results. The RF algorithm successfully generated detailed maps of gamma-spectrometric and κ variables. The distribution of radionuclides and ferrimagnetic minerals was influenced by morphometric variables. Nested-LOOCV method evaluated algorithm performance accurately with limited samples, generating robust mean maps. The highest thorium levels were observed in elevated, flat, and west beach areas, where detrital materials from periglacial erosion came





through fluvioglacial channels. Lithology and pedogeomorphological processes-controlled thorium contents. Steeper areas
formed a ring with the highest uranium contents, influenced by lithology and geomorphological-periglacial processes (rock
cryoclasty, periglacial erosion, and heterogeneous deposition). Felsic rocks and areas less affected by periglacial erosion had
the highest potassium levels, while regions with sulfurization-affected pyritized-andesites near fluvioglacial channels showed
the lowest potassium contents. Lithology and pedogeochemical processes governed potassium levels. The κ values showed no
distinct distribution pattern. Pyritized-andesite areas had the highest levels due to sulfurization and associated pyrrhotite,
promoting iron release. Conversely, Cryosol areas, experiencing freezing and thawing activity, had the lowest κ values due to
limited ferrimagnetic mineral formation. Lithology and pedological-periglacial processes in Cryosols played a significant role
in controlling κ values. In regions characterized by diverse terrain attributes and abundant active and intense periglacial
processes, the spatial distribution of geophysical variables does not reliably reflect the actual lithological composition of the
substrate. The complex interplay of various periglacial processes in the area, along with the morphometric features of the
landscape, leads to the redistribution, mixing, and homogenization of surface materials, contributing to the inaccuracies in the
predicted-spatialized geophysical variables.

**Keywords**: modeling; cryosphere; geophysical characterization; geoscience

## 1. Introduction

Recently, proximal geophysical sensors have been used for lithological-pedological characterization, in other parts of the
world, where the provided information is used to understand the lithosphere-pedosphere interaction and dynamics in a
geoenvironmental context (Bastos et al., 2023; Vingiani et al., 2022). Geophysical surveys utilizing these sensors enabled the
gathering of field information swiftly and efficiently, eliminating the necessity of sample collection for laboratory analysis
(Souza et al., 2021; Mello et al., 2020; Mello et al., 2021; Mello et al., 2023; Mello et al., 2023). Among the geophysical
techniques used for lithological surveys, radiometric (gamma-ray spectrometry) and magnetic (magnetic susceptibility) stand
out.

Gamma-spectrometry involves the quantification uranium ($e^{238}U$), thorium ($e^{232}Th$), and potassium ($^{40}K$) commonly called
(radionuclides) in naturally radioactive rocks, soils, and sediments (Minty, 1988). The quantities of these radionuclides are
influenced by various factors such as lithological substrate and surface processes (mainly weathering, pedogenesis,
pedogeomorphological, and periglacial) (Navas et al., 2018). Past studies undertaken by Dickson and Scott, (1997); Wilford
and Minty, (2006) and Mello et al., (2021) have shown that the radionuclide contents are dependent on these factors. Proximal
gamma-ray spectrometry, as highlighted by (Ford et al., 2008), offers a precise method for determining concentrations of
specific radioactive elements and mapping their sources accurately in soil, bedrock, and surface geological exploration.

Magnetic susceptibility (κ) is a measure of the magnetization induced in a sample relative to the magnetic field inducing it
(Mullins, 1977). It is influenced by the presence of ferrimagnetic primary minerals in both the lithological substrates and the
ferrimagnetic secondary minerals found in the sand (magnetite) and clay (maghemite) fraction of the soil (Ayoubi et al., 2018).





In addition, the surface processes a role in determining κ values (Garankina et al., 2022; Mello et al., 2020; Ribeiro et al., 2018; Sarmast et al., 2017).

Many studies used gamma-ray spectrometry mapping to delineate lithological maps (Arivazhagan et al., 2022; Loiseau et al., 2020; Shebl et al., 2021) and magnetic susceptibility (Bressan et al., 2020; Costa et al., 2019; Harris and Grunsky, 2015). In addition, recently Mello et al., (2022), Mello et al., (2020), Mello et al., (2021), Mello et al., (2022), Mello et al., (2022) have successfully used machine learning algorithms combined with data from multiple field geophysical equipment to map geophysical variables and understand tropical soils, lithology and landscapes, obtaining satisfactory results in mapping and understanding these landscapes using modeling via machine learning algorithms.

While geophysical survey techniques are well-established and commonly utilized in research, only a few studies have demonstrated their use, specifically gamma-ray spectrometry and magnetic susceptibility, for characterizing and understanding periglacial landscapes (Mello et al., 2022). This scarcity of studies is particularly evident in the Antarctic environment, known for its hostile and inaccessible nature, where there is a lack of geophysical characterization, such as gamma-spectrometric or magnetic susceptibility mapping, and their correlation with periglacial processes and landforms. Fieldwork logistics under such unique geoenvironmental conditions are challenging. Therefore, acquiring geophysical data in the field could facilitate more precise deductions about the lithological, mineralogical, and pedological traits of an area, ultimately reducing the necessity for physical sample collection and laboratory analyses. This advantage is especially crucial for research in this domain, particularly when employing different sensor technologies.

Maritime Antarctica (MA) is currently a great geosciences frontier to be explored with its complex and heterogeneous landforms and lithological characteristics. MA has a different climate from continental Antarctica, being hotter and more humid (Turner et al., 2007, 2005). In this region, periglacial environments are abundant and ruled by seasonal cycles of water freezing-thawing, which determine the specific landforms, permafrost and typical soils (French, 2017; Pollard, 2018). The MA lithology is predominantly composed of igneous rocks and a few sedimentary rocks. This complex lithological system associated with climatic conditions produces heterogeneous soils, sediments and saprolites, forming a unique geoenvironment on the planet (Meier et al., 2023; Siqueira et al., 2022).

Traditionally geoscientists use invasive, destructive and time-consuming techniques for lithological and pedological characterization in natural systems, employing sample collection for physical-chemical and mineralogical analysis in the laboratory. Besides, the lack of detailed characterization of samples in the field demands a high collection of samples. In Antarctica, material collection is limited by a lack of logistics and restricted access to a small number of researchers who sample on the continent.

Given the above, we conducted research that aimed: *i)* to perform a detailed lithological-pedological radiometric (ternary map) and κ surficial surveys in Keller Peninsula (MA), using machine learning and Nested-leave one-out cross validation method and; *ii)* to explore the relationship between geophysical variables and periglacial processes across a heterogeneous igneous rock and landscape. Specifically, we: 1) characterized and mapped the distribution of surficial natural radioactivity (from $U^{238}$, $^{232}Th$ and $^{40}K$ radionuclides) and magnetic susceptibility (κ) to produce a ternary gamma-ray and a κ maps; 2) Tested the



efficiency of a machine learning algorithm in predicting and spatializing geophysical data using the nested-leave-one-out cross validation method; 3) relate the distribution of natural radioactivity and κ to pedogeomorphological and periglacial processes and dynamics in the different lithological-pedological substrates and outcrop rocks.

This study can improve our understanding about periglacial processes, which can improve geophysical surveys and soil-lithological digital mapping in the Antarctic environment. This expectation is based on research that has focused on

comprehending lithological characteristics, periglacial processes and landscape evolution in Antarctic pedosphere-lithosphere interactions.

## 2 Material and methods

### 2.1 Study area, lithological-soil surveys and sampling points

The study site comprises Keller Peninsula (62°4′33″ S, 58°23′46″W), Admiralty Bay, King George Island, and in the South Shetland Archipelago in M.A. (**Fig. 1**). The Keller Peninsula covers an area of 500 ha, stretching 4 km (north-south) and 2 km (east-west) (Francelino et al., 2011).




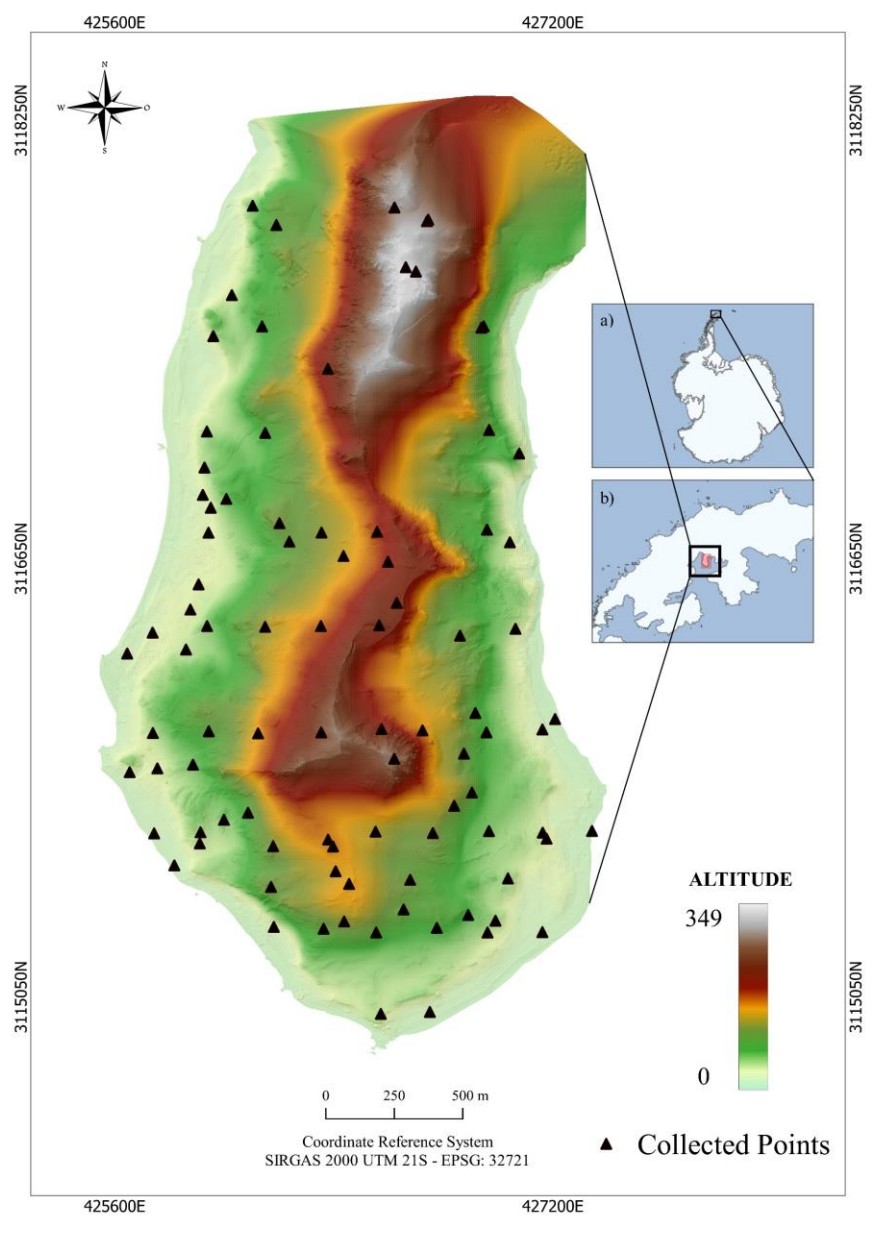

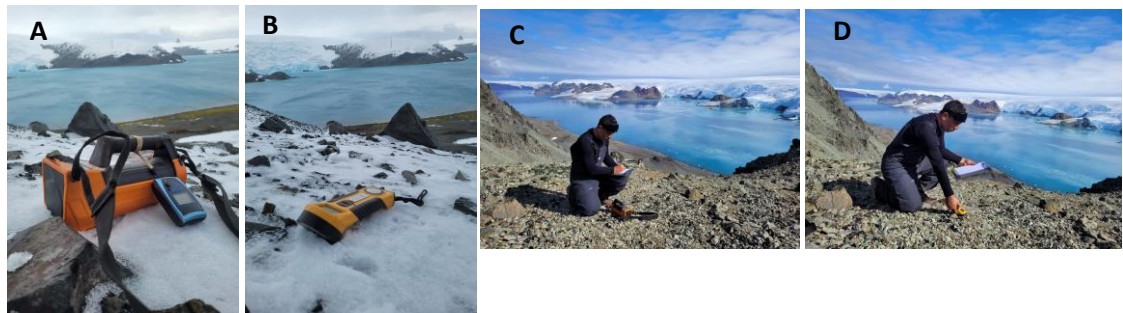





**Figure 1.** Study area in Maritime Antarctic (Keller Peninsula), collected point, digital elevation model, and geophysical
sensors. a) Antarctic continent; b) Keller Peninsula. A – Gamma-ray spectrometer (Radiation Solution – RS 230); B – Magnetic
susceptibility meter (KT-10 Terraplus); C – Gamma-ray readings; D- Magnetic susceptibility readings.

The weather in Maritime Antarctica follows a typical pattern, albeit slightly warmer, as outlined by Rakusa-Suszczewski et
al., (1993). Summer (December–March) temperatures average around +1.6°C, whereas winter (June–September) temperatures
drop to an average of −5.3 °C (INPE, 2009). Annual precipitation is around 400 mm. The Keller Peninsula spans elevations
between 0 and 380 meters above sea level, featuring a diverse topography from flat to steep (slopes ranging from 0 to 75%).
This region is characterized by various landforms like moraines, protalus, inactive rock glaciers, uplifted marine terraces, and
Felsenmeer. These geological features have formed due to both paraglacial and periglacial conditions, as discussed by
Francelino et al., (2011).
Between 1948 and 1960, British geologists conducted a detailed geological-lithological map of Admiralty Bay and its
surrounding area with a scale of 1:50,000 (Birkenmajer, 1980), which was used in this research. The predominant lithological
composition of the Keller Peninsula consists mainly of igneous rocks, specifically andesitic-basalts, basaltic-andesites, diorite,
pyritized-andesite, tuffites, and unspecified marine sediments, as depicted in Figure 2A. The lower regions of the landscape
host marine terraces, whereas the upper regions are primarily characterized by andesitic-basalts and basaltic-andesites.
Scattered across the area are the pyritized-andesites, with tuffs occupying intermediate altitudes. Diorites, though less
prevalent, are localized to specific areas.

**Figure 2**. The maps indicated: A) Lithology. B) Soil classes: The lithological and soil classes maps were adapted from Francelino et al., (2011).


Soil classification and mapping were carried out by an experienced pedologist, using 20 representative soil profiles. The overall soil development in the area is limited, and according to the World Reference Base for Soil Resources (WRB) (FAO, 2014), 2014), the soils in the region can be categorized into various types, including Gelic Eutric Leptosol, Gelic Skeletic Regosol, Gelic Skeletic Cambisol, Gelic Leptic Regosol, Gelic Dystric Fluvisol, Arenic Skeletic Cryosol, Vitric Leptic Cryosol, Gelic

Skeletic Regosol, Gelic Leptic Cambisol, and Arenic Turbic Cryosol, as illustrated in Figure 2B. The occurrence of permafrost was observed in just five soil profiles, situated below a depth of 2 meters in coastal areas. Additionally, it is discontinuously found within the first two meters in regions with mid-slope and flat topography, all of which are classified as Cryosols (Francelino et al., 2011; Mello et al., 2023). These permafrost occurrences were observed in only five soil profiles, all of which are classified as Cryosols. Within each soil profile, samples were meticulously collected from identified diagnostic soil

horizons and at various soil depths to facilitate subsequent physico-chemical analyses.





The sampling design and the selection of measurement locations were conducted while considering several toposequences (topographic gradients) that account for various lithologies and soils. The distribution of collection points by the proximal gamma-ray and magnetic susceptibilimeter is shown in **figure 1**. The readings with the sensors were carried out, taking into account the lithological diversity of the Peninsula, as well as pedological diversity and variations in relief.

## 2.2 Geophysical survey, radiometric and κ characterization

The geophysical variables (radiometric and κ), were collected using proximal geophysical sensors, RS-230 and KT-10 Terraplus, respectively (Figs. 1A and B, respectively). The radiometric data (gamma-ray spectrometry) correspond to the acquisition of radionuclide contents eU, eTh and $^{40}$K, quantified in ppm (eU and eTh) and % for $^{40}$K. Magnetic susceptibility is given in 10-3 SI units and the sensor is able to detect mean κ values to a depth of 2 cm below the rock outcrops surface (Sales, 2021). Detailed calibration methods, method of collection and interpretation of results can be found in (Mello et al., 2022; Mello et al., 2021, 2022).

Gamma spectrometric readings (**Fig. 1 C**) were taken on the rock outcrop surface and soil depth (saprolite/rock), at the 91 collection points shown in figure 1, in "essay mode", which provides greater precision, and the reading time was adjusted to 3 minutes at each point. The sensor is able to detect radiation up to a mean depth of 30 - 60cm depending on the characteristics of the substrate (Beamish, 2015; Taylor et al., 2002; Wilford et al., 1997a). Subsequently, the equipment data were transferred to a computer and concatenated with soil κ values and environmental data (lithology and terrain attributes).

Similarly, the κ survey readings (**Fig. 1 D**) was undertaken at each of the 87 points to a mean depth of 2 cm below the rock outcrops surface and soil depth (saprolite/rock). Three readings were taken for each point to reduce sensor noise and increase the precision reading and, the κ mean values of these three readings were used in data processing. The resulting κ data were then combined with their respective gamma-ray spectrometric data, lithology and terrain attributes in order to be processed.

All readings were carried out in different lithological-pedological compartments, with emphasis on in situ materials (igneous rocks), despite of the little presence of marine terraces with presence of external materials and a wide range of materials from ex situ sites.

## 2.3 Digital Elevation Model

Geoprocessing and Digital Elevation Model (DEM) analysis were conducted utilizing R software version 4.10 version (R Core Team, 2023), employing data derived from a High-Resolution Topography (HRT) survey to create the DEM (**Fig. 1**). The HRT survey, conducted during the 2014/2015 and 2015/2016 periods, utilized a Terrestrial Laser Scanner (TLS) of the RIEGL VZ-1000 model, known for its nominal accuracy and precision of 8 and 5 mm, respectively (Schünemann et al., 2018). This advanced sensor and geoprocessing methodology yielded a low root mean square error and a high number of points per cell, resulting in a densely populated point cloud. This dataset facilitated a comprehensive generalization process to generate surface models with superior performance, accurately representing local relief. This, in turn, enabled in-depth studies of landscape evolution at a micro scale over time, specifically allowing for the assessment of pedogeomorphological processes.





Using the R software (R Core Team, 2023), a total of forty-eight additional topographic attributes were computed based on
the DEM data extracted from the Digital Terrain Model (DTM) (**Table 1**). These attributes were derived through the utilization
of the "Rsaga" tool (Brenning, 2008) and the "raster" package (Hijmans and Van Etten, 2016).

**Table 1.** Terrain attributes generated from the digital terrain model, geology, soil and spectral indices.

| Terrain attributes, geology and spectral indices | Abbreviations | Brief description |
|---|---|---|
| **Aspect** | AS | Slope orientation |
| **Blue Band** | B | The blue band wavelengths fall below 1546.12 nm. |
| **Green Band** | G | BA primary wavelength of 495–570 nm approximately |
| **Red Band** | R | The longer wavelengths of 1546.12 nm and higher |
| **Curvature classification** | CC | Curvature classification |
| **Convergence index** | CI | terrain parameter which shows the structure of the relief as a set of convergent areas (channels) and divergent areas (ridges). |
| **Difference** | D | Geometric difference of the overlayed polygon layers |
| **Diurnal anisotropic heating** | DAH | Continuous measurement of exposure dependent energy |
| **Easterners** | EA | |
| **Flow Line Curvature** | FLC | Represents the projection of a gradient line onto a horizontal plane |
| **General curvature** | GC | The combination of both plan and profile curvatures |
| **Geology** | GEO | Rocks and similar substances that make up the earth's surface |
| **Hill shade** | HI | A technique where a lighting effect is added to a map based on elevation variations within the landscape. |
| **Digital elevation model** | DEM | Representation of the bare ground (bare earth) topographic surface of the Earth excluding trees, buildings etc. |
| **Effective air flow heights** | EAFH | A line representing the resultant velocity of the disturbed airflow |
| **Longitudinal curvature** | LC | Measures the curvature in the down slope direction |
| **Mass balance index** | MBI | Multivariate distance methods for geomorphographic relief classification |
| **Maximal curvature** | MAXC | Maximum curvature in local normal section |
| **Mid-slope position** | MSP | Represents the distance from the top to the valley, ranging from 0 to 1 |
| **Minimal curvature** | MINC | Minimum curvature for local normal section |




| Terrain attributes, geology and spectral indices | Abbreviations | Brief description |
| --- | --- | --- |
| Morphometric Protection Index | MPI | Measure of exposure/protection of a point from the surrounding relief |
| Multiresolution index of ridge top flatness | MRRTF | Indicates flat positions in high altitude areas |
| Multiresolution index of valley bottom flatness | MRVBF | Indicates flat surfaces at the bottom of valley |
| Normalized Difference Vegetation Index | NDVI | Remote sensing technique used to assess the health and density of vegetation. |
| Normalized height | NH | Vertical distance between base and ridge of normalized slope |
| Northerns | NO | Means in or from the north of a region |
| Ridge level | RL | The maximum vertical distance between the finished floor level and the finished roof height directly above. |
| Slope | S | Represents local angular slope |
| Slope height | SH | Vertical distance between base and ridge of slope |
| Slope Index | SI | Represents a local angular slope index |
| Solrad Diffuse1 | SolDiffuse1 | Diffuse insolation for the month of January |
| Solrad Diffuse2 | SolDiffuse2 | Diffuse insolation for the month of July |
| Solar total radiation | SolTR | Insolation duration for the month of January |
| Solrad Direct1 | SolDiret1 | Direct insolation for the month of January |
| Solrad Direct2 | SolDiret2 | Direct insolation for the month of July |
| Solrad Ration1 | SolRation1 | Ratio between direct insolation and diffuse insolation for the month of January |
| Solrad Ration2 | SolRation2 | Ratio between direct insolation and diffuse insolation for the month of July |
| Soil | S | Soil body as triphasic system |
| Sky view factor | SVF | Defines the ratio of sky hemisphere visible from the ground |
| Standardized height | STANH | Vertical distance between base and standardized slope index |
| Tangential curvature | TANC | Measured in the normal plane in a direction perpendicular to the gradient |
| Terrain ruggedness index | TRI | Quantitative index of topography heterogeneity |
| Terrain surface convexity | TSC | Ratio of the number of cells that have positive curvature to the number of all valid cells within a specified search radius |





| Terrain attributes, geology and spectral indices | Abbreviations | Brief description |
|---|---|---|
| **Terrain surface texture** | TST | Splits surface texture into 8, 12, or 16 classes |
| **Total curvature** | TC | General measure of surface curvature |
| **Total insolation** | TSR | The amount of solar energy that strikes a given area over a specific time, |
| **Topographic openness** | TO | Expresses the dominance (positive) or enclosure (negative) of a landscape location. |
| **Topographic position index** | TPI | Difference between a point's elevation and surrounding elevation |
| **Valley depth** | VD | Calculation of vertical distance at drainage base level |
| **Valley** | VA | Calculation of fuzzy valley using the Top Hat approach |
| **Valley Index** | VA | Calculation of fuzzy valley index using the Top Hat approach |
| **Vector ruggedness index** | VRI | Measure terrain ruggedness as the variation in three-dimensional orientation |
| **Topographic wetness index** | TWI | Describes the tendency of each cell to accumulate water as a function of relief |
| **Wind exposition** | WE | The average of wind effect index for all directions using an angular step |

## 2.4 Modeling processes and statistical analysis

The point values of eU, eTh, $K^{40}$, and κ, linked with terrain attributes, soil type, lithology, and RGB (**Table 1**), were utilized to modeling these variables for other areas, employing the Random Forest (RF) algorithm. The modeling process comprises two main steps: covariate selection and model tuning / performance evaluation. During the covariates the selection phase, the algorithm aims to generate an optimal set of covariates, adhering to the principle of parsimony. This involved two sequential methods, we initially removed self-correlated variables and subsequently assessed the importance of the remaining variables. Initially, to assess the correlation between variables, we used a Spearman correlation cut-off limit > |0.95|. We eliminated a variable from the pair of variables with correlation above the established value, to decide which variable from the pair would be eliminated, we opted for the variable with the highest sum of absolute correlations with the other covariates involved in this process. To carry out this phase, we employed the "cor" and "findcorrelation" functions from the "stats" (Hothorn, 2021) and "caret" (Kuhn et al., 2020) packages in the R software, respectively (Kuhn and Johnson, 2013). The covariates that successfully passed this selection phase were combined with the samples and, subsequently, the samples were separated into training and test sets.

To partition the data into training and test subsets, we adopted the "nested leave-one-out cross-validation" (nested LOOCV) method (Ferreira et al., 2021; Paes et al., 2022; Rytky et al., 2020). It is noteworthy that the number of samples and readings





obtained from geophysical sensors was limited (92) due to various challenges encountered during data collection in the field (e.g., sloping terrain, high hazard areas, glaciers, steep terrain, snowbanks, etc.). Given the small sample size, the nested LOOCV method was chosen, as this method has already been recommended by other authors in similar cases (Ferreira et al., 2021; Mello et al., 2022a; Mello et al., 2022c, 2022b).  This particular approach represents a significant innovation in our research.

The nested LOOCV approach involves a double looping process. In the first loop, the model is trained on a dataset of size n-1, and in the second loop, testing is performed using the omitted sample to evaluate the training performance (Jung et al., 2020; Neogi and Dauwels, 2019). The final machine learning algorithm performance results are calculated as average performance indicators across all points (training/testing). This method proves to be robust in evaluating the real generalization ability of the algorithm and in identifying possible problematic samples or outliers in the data set. Each iteration generates a training set

that undergoes covariate selection by importance and subsequent training.

The covariate selection based on importance is executed using the backward-forward method, employing the Recursive Feature Elimination (RFE) function available in the "caret" package (Kuhn and Johnson, 2013). This RFE technique is algorithm-specific and yields an optimal set of covariates utilized in predicting the final model for each respective algorithm. RFE is a selection procedure that iteratively removes variables contributing the least to the model, employing an importance measure

tailored to each algorithm (Kuhn and Johnson, 2013).

The algorithm is then trained on discrete subsets of variables, going from 2 to the total variables one by one. The ideal subset of covariates is optimized based on the leave-one-out cross-validation (LOOCV), for each of the internal hyperparameters of the tested algorithms (10). The hyperparameters for each algorithm are described in the caret package manual, Chapter 6, "Described Models", available at https://topepo.github.io/caret/train-models-by-tag.html. The Mean Absolute Error (MAE)

was used as a metric to select the best subset for the RF algorithm.

Training is then performed using the previously selected variables and tested with LOOCV. Additionally, ten values of each RF hyperparameter were evaluated. At the end of the training phase, predictions are made on samples not used in the training process, and the results are saved for performance analysis. The assessment of algorithm predictions and sensor sets is carried out using a collection of samples from the outer loop within the nested Leave-One-Out Cross-Validation (LOOCV) method.

Three key evaluation parameters are utilized: Concordance Correlation Coefficient (CCC) (Eq. (1)), Root Mean Square Error (RMSE) (Eq. (2)), and Mean Absolute Error (MAE) (Eq. (3)) (Lin, 1989).

$$\rho_c = \frac{2p\sigma_x\sigma_y}{\sigma_x^2 + \sigma_y^2 + \left(\mu_x - \mu_y\right)^2} \tag{1}$$

Where:

n represents the number of samples;

$\rho_c$ is the correlation coefficient between the two variables;

$\mu_x$ and $\mu_y$ are the means for the two variables;





"σ" _"x" ^"2" and "σ" _"y" ^"2" are the corresponding variances;

Pi and Oi represent the predicted and observed values at location i.

$$RMSE = \sqrt{\frac{1}{n} \times \sum (Qobs - Qpred)^2} \qquad (2)$$

$$MAE = \frac{1}{n} \times \sum |Qpred - Qobs| \qquad (3)$$


Where:

Qpred = the mean of the training samples

Qobsi = the validation sample

n = number of samples (loop).


As additional validation, we used the "null model" approach (NULL_RMSE and NULL_MAE). This null model involves using the mean value determined from the collected samples (EQ. 4 and EQ. 5). The null model represents the simplest possible model when given a training set, providing a single average value for numerical results.

The null model serves as a reference and can be seen as the simplest adjustable model. Any other models that present similar

or inferior performances in relation to the null model must be discarded. This indicates that the final model outperforms the use of average values, highlighting its superior quality in model creation. Furthermore, the null model approach is widely employed, especially in spatialization processes such as kriging, where the variable under consideration exhibits spatial dependence, often called the pure nugget effect. The equations used for NULL_RMSE and NULL_MAE calculations are as follows:


$$NULL\_RMSE = \left[\frac{1}{N}\sum_{i=1}^{N}(\overline{Qtrain}_i - Qobs_i)^2\right]^{\frac{1}{2}} \qquad (Eq.4)$$

$$NULL\_MAE = \frac{1}{n} \times \sum|\overline{Qtrain}_i - Qobs_i| \qquad (Eq.5)$$

Where:

Qtrain = the mean of the training samples

Qobsi = the validation sample

n = number of samples (loop).



The NULL_RMSE and NULL_MAE values were computed using the nullMode function within the caret package (Kuhn et al., 2020). To assess the overall performance of the algorithms for each attribute, a total of 87 loops were conducted. The training results represent the average performance, and the test sample results were calculated from the 87 outer loops using Equations 1, 2, and 3.

Eighty-seven maps were predicted, yielding one map for each execution of the outer loop in the nested Leave-One-Out Cross-
Validation (LOOCV). Coefficient of Variation (CV) was calculated for each pixel across the 87 stacked maps. Additionally, a coefficient of variation map (CV% = standard deviation / mean) was generated to illustrate the variation of predicted values in each pixel of the map relative to the mean. Spatial predictions exhibiting lower CV indicate more consistent results, thereby resulting in smaller errors in the estimation/predictions and reduced uncertainty.

The statistical differences between the geophysical variables and lithology substrates were analysed by using the Kruskal-
Wallis and Dunn's posthoc tests with a significance level of 5%.

It is important to highlight that only 87 points with geophysical sensors were taken on pedological substrates. Furthermore, these few points are found in soils with little pedogenetic evolution, characterized by a high content of rock fragments and a predominance of the coarse fraction composed of cryoclastic rocks (many with a skeletal character). Additionally, where there was soil, we opened a small trench and carried out geophysical readings at the base of the soil profiles, in direct contact with
the rock. As a result, we do not have enough number of points to carry out an analysis to identify differences between surface geophysical variables and pedological substrates. Therefore, we consider these points as readings carried out on the lithological substrate.

## 3 Results and discussion

### 3.1. Model's performance and variable's importance

The RF algorithm was used to predict gamma-ray data and magnetic susceptibility of the substrate to produce ternary gamma-ray and κ maps (Table 2). The CCC values ranges from 0.771 to 0.851 (Table 2). The importance of covariates to geophysical variables prediction showed that morphometric (minimal curvature, mid-slope position, diurnal anisotropic heating, difference, total insolation, flow line curvature, effective air flow heights, terrain surface convexity, hill shading, aspect, mass balance
index, ridge level, digital elevation model, and convergence index), were the most important variables, contributing more than 50% on the decreasing of the mean accuracy (**Fig. 3**). On the other hand, lithology contributed little to the prediction of geophysical variables (less than 50%) (**Fig. 3**). Cracknell and Reading, (2014), Harris and Grunsky, (2015) and Kuhn et al., (2018), also reached satisfactory performance using RF algorithm to predict radionuclides contents and magnetic susceptibility to perform lithological mapping.




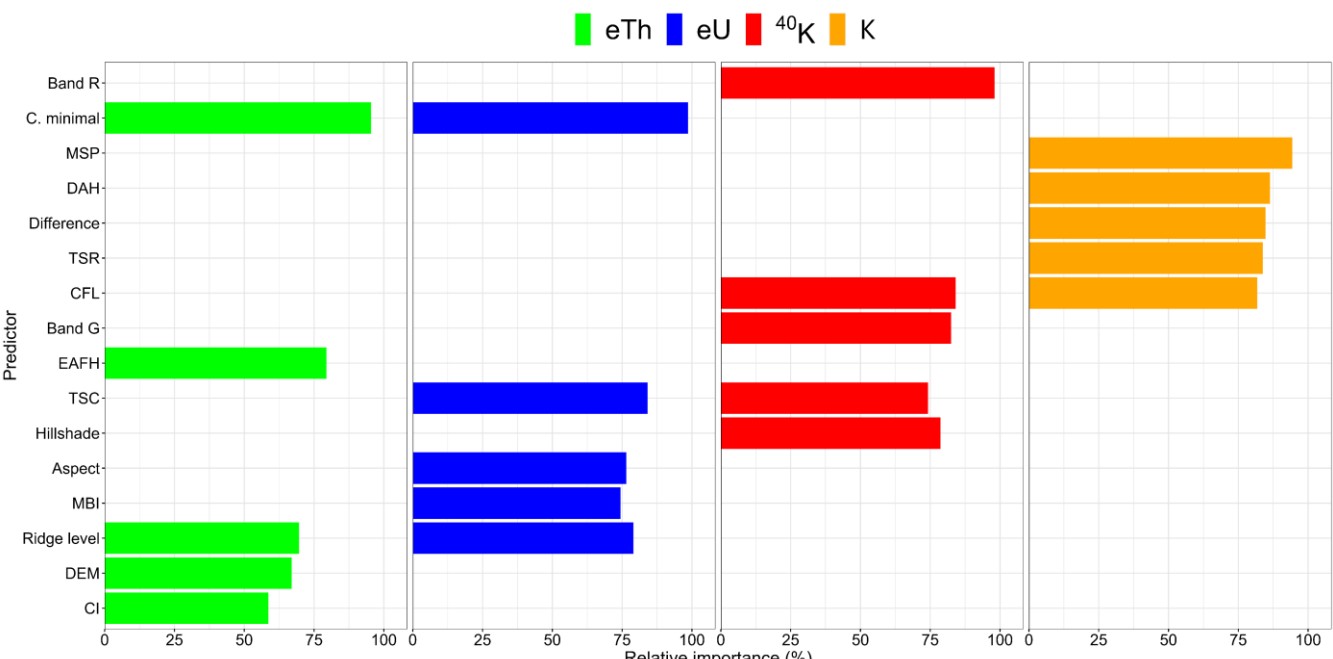

**Figure 3.** Importance of variables of predictors. *X* axis: variables that most contributed to the predictive models. Axis *y*: value in relative percentage of the contribution. eTh: equivalent thorium; eU: equivalent uranium; [40]K potassium; κ: magnetic susceptibility.


**Table 2.** Model's performance in terms of Concordance Correlation Coefficient (CCC), Root Mean Squared Error (RMSE), Mean Absolute Error (MAE), *NULL_RMSE* and *NULL_MAE*.

| Parameters of model's performance | RF Algorithm | | | |
|---|---|---|---|---|
| | eU | eTh | [40]K | κ |
| *CCC* | 0.771 | 0.851 | 0.817 | 0.809 |
| *MAE* | 0.496 | 1.784 | 0.497 | 11.898 |
| *RMSE* | 0.680 | 2.432 | 0.646 | 15.469 |
| *NULL_RMSE* | 0.652 | 2.457 | 0.502 | 12.224 |
| *NULL_MAE* | 0.463 | 1.818 | 0.502 | 15.651 |

Concordance Correlation Coefficient (CCC) is a modified version of the coefficient of determination ($R^2$). Apart from measuring the correlation strength, CCC also indicates how closely the model predictions align with a 45-degree inclined line (line 1:1) from the origin. This characteristic makes it an effective tool to assess both the precision and accuracy of forecasts





(Svensson et al., 2022; Zhao et al., 2022). Unlike the Pearson correlation coefficient, the CCC can detect bias in the model predictions. This difference between the two metrics makes the CCC a more suitable choice for validation compared to the coefficient of determination (Khaledian and Miller, 2020). Recent researches in geosciences have satisfactorily used the CCC indices as a parameter for evaluating the performance of machine learning algorithms (Chen et al., 2019; S. Chen et al., 2019; Feng et al., 2019; Khosravi et al., 2018; Mishra et al., 2022; Siqueira et al., 2023; Zhou et al., 2022).

Practically all the relief variables are associated with the landform that control the surface periglacial and pedogeomorphological processes of the Keller Peninsula landscape. Periglacial erosion, glacial fluvial melt channels, freezing and thawing of the active layer of permafrost and solifluxion are the most frequent periglacial and pedogeomorphological processes in Keller Peninsula, as observed by Francelino et al., (2011) and López-Martínez et al., (2012). These processes promote the fragmentation, redistribution and mixing of materials in significant areas of the landscape, which can contribute to variations in radionuclide and k values, as well as increase prediction errors in the points of greater occurrence of these processes, such as the sloping areas of the landscape.

The nested-LOOCV methodological framework was better than NULL-model, for prediction of radionuclides and magnetic susceptibility with a limited number of samples. This approach consistently generated comparable maps across loops, where 87 samples were utilized for training in each loop, and at the conclusion of the process, all samples were used as the test dataset. As a result, the models and/or coefficients of variation in the maps were similar (Ferreira et al., 2020).

## 3.2 Radionuclides and κ contents on lithological compartments and their relationship with mineralogy

The eU mean content was generally low and showed the greatest variation on the lithologies (**Fig. 4**). The highest eU mean content values were observed on tuffites and the lowest on andesitic-basalts (**Fig. 4**). The diorite presented the highest mean eTh contents, while the andesitic-basalts showed the lowest one. Regarding the $^{40}$K, the mean values were high in all lithologies (> 1%) excepted on pyritized-andesite (**Fig. 4**). The highest $^{40}$K contents were observed on diorite, and the lowest one on pyritized-andesite (**Fig. 4**). The mean κ values ranged from moderate to low in all lithologies, where pyritized-andesites showed the highest mean values and tuffites the lowest (**Fig. 4**). The descriptive statistics for radionuclides and κ content for all lithological units are shown in table 3, and corroborates and complements the information provided in figure 4 in quantitative terms.





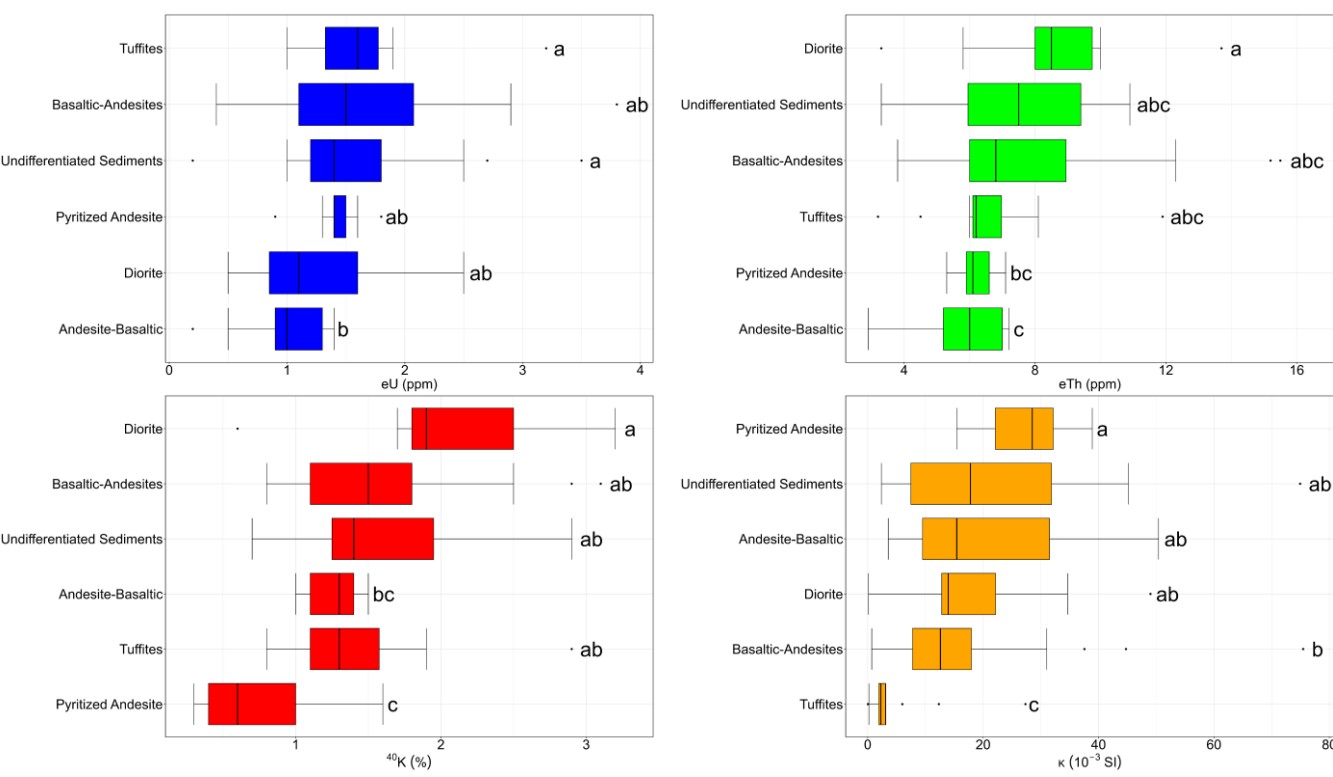

**Figure 4.** *Boxplot* with descriptive statistics of the distribution of radionuclide contents and magnetic susceptibility by lithology. Lowercase letters to the right of the boxplot bars indicate statistical differences as determined by the Kruskal-Walli's test.








**Table 3. Descriptive statistics for the analyzed radionuclides and κ by lithology**


### Diorites

| Summary Statistics | Radiometric and magnetic susceptibility | | | |
|---|---|---|---|---|
| | $eU(mg.kg^{-1})$ | $eTh(mg.kg^{-1})$ | $^{40}K(\%)$ | $\kappa\ (10^{-3}\ SI\ units)$ |
| Mean | 1.27 | 8.85 | 2.10 | 18.54 |
| Standard deviation | 0.62 | 3.01 | 0.74 | 13.72 |
| Minimum | 0.50 | 3.30 | 0.60 | 0.13 |
| Maximum | 2.50 | 13.7 | 3.20 | 49.0 |

### Tuffites

| Summary Statistics | Radiometric and magnetic susceptibility | | | |
|---|---|---|---|---|
| | $eU(mg.kg^{-1})$ | $eTh(mg.kg^{-1})$ | $^{40}K(\%)$ | $\kappa\ (10^{-3}\ SI\ units)$ |
| Mean | 1.63 | 6.62 | 1.41 | 4.73 |
| Standard deviation | 0.55 | 1.96 | 0.53 | 7.18 |
| Minimum | 1.00 | 3.20 | 0.80 | 0.04 |
| Maximum | 3.20 | 11.9 | 2.90 | 27.33 |

### Andesitic-basalts

| Summary Statistics | Radiometric and magnetic susceptibility | | | |
|---|---|---|---|---|
| | $eU(mg.kg^{-1})$ | $eTh(mg.kg^{-1})$ | $^{40}K(\%)$ | $\kappa\ (10^{-3}\ SI\ units)$ |
| Mean | 0.99 | 5.72 | 1.26 | 21.0 |
| Standard deviation | 0.41 | 1.44 | 0.17 | 15.0 |
| Minimum | 0.20 | 2.90 | 1.00 | 3.56 |
| Maximum | 1.40 | 7.20 | 1.50 | 50.4 |

### Basaltic-andesites

| Summary Statistics | Radiometric and magnetic susceptibility | | | |
|---|---|---|---|---|
| | $eU(mg.kg^{-1})$ | $eTh(mg.kg^{-1})$ | $^{40}K(\%)$ | $\kappa\ (10^{-3}\ SI\ units)$ |
| Mean | 1.62 | 7.87 | 1.59 | 17.2 |
| Standard deviation | 0.82 | 3.08 | 0.61 | 16.6 |
| Minimum | 0.40 | 3.80 | 0.80 | 0.70 |
| Maximum | 3.80 | 15.5 | 3.10 | 75.4 |

### Pyritized-andesites

| Summary Statistics | Radiometric and magnetic susceptibility | | | |
|---|---|---|---|---|
| | $eU(mg.kg^{-1})$ | $eTh(mg.kg^{-1})$ | $^{40}K(\%)$ | $\kappa\ (10^{-3}\ SI\ units)$ |
| Mean | 1.64 | 7.59 | 1.54 | 22.5 |
| Standard deviation | 0.74 | 2.23 | 0.57 | 18.6 |
| Minimum | 0.20 | 3.30 | 0.70 | 2.37 |
| Maximum | 3.50 | 10.9 | 2.90 | 74.9 |





In our study site, the great variability in eU content and/or their low content in basic-intermediate igneous rocks (**Fig. 4, table 3**) can be explained mainly by two reasons: i) loss of uranium from magmas during the later stages of their differentiation; ii)

uranium oxidation and reduction in its content with increasing in fractionation processes during magmatic crystallization (Ragland et al., 1967; Whitfield et al., 1959). In addition, according to Wilford and Minty, (2006) and Wilford et al., (1997b), the mean contents of radionuclides in the Earth crust vary to 2.3%, 3 ppm and 12 ppm, for $^{40}K$, uranium and thorium, respectively, similar to the values observed in our study site.

The eTh and $^{40}K$ contents tended to increase with an increase in the silicon content in our lithology composed by igneous rocks

(from andesitic-basalts to diorite) (**Fig. 4, table 3**). Our results are corroborated by Dickson and Scott, (1997) and Mello et al., (2023b), who found an increasing in eTh and $^{40}K$ contents in acid-felsic igneous rocks and lower levels in basic-mafic igneous rocks. It is noteworthy that the undifferentiated sediments receive materials from various parts of the Peninsula and from outside it, in which case it is not appropriate to use this lithology for radionuclides comparison purposes. Most of the gamma-ray radiation detected and quantified by the sensor originates from the first 45-60cm of the dry substrate (rocks, soils and

sediments), which the mineralogy and geochemistry of the substrate presented the greater contribution to radionuclides contents (Gregory and Horwood, 1961). In addition, Earth surface processes and landforms such as chemical weathering, pedogenesis and relief affect radionuclide contents, since $^{40}K$ content decreases with weathering advance once it is removed by destruction of feldspars. Also, $^{40}K$ is not incorporated in secondary minerals, so it is leached, whereas Th composes highly resistant minerals, such as ilmenite and zircon. Consequently, Th content increases with weathering (Dickson and Scott, 1997;

Wilford et al., 1997b; Mello et al., 2021; Mello et al., 2022; Mello et al., 2023). In our case, gamma radiation and radionuclide contents are much more associated with the mineralogy of the substrate than land surface processes, due to the low intensity with which chemical weathering and pedogenesis operate in MA. Despite this, physical weathering in this environment associated with periglacial processes (governed by cycles of freezing and thawing of water in the different portions and types of substrates) should not be neglected, since during these processes, radionuclides are redistributed in the landscape.

The low mean κ values were not expected on basic mafic igneous lithologies (basaltic-andesite, andesitic-basalts) (**Fig. 4, table 3**), since there is a great probability to these rocks present more abundance of ferrimagnetic minerals in the rock. According to Mullins (1977) increasing in ferrimagnetic mineral contents in the substrate results in increases in κ values. The low κ values associated with the low content of ferrimagnetic minerals in the basaltic-andesite and andesitic-basalts may be due to a difficulty in stabilizing the iron oxides mineral phases caused by low pressures, low oxygen content and anhydrous conditions

in magma source. This would reduce the iron oxides content in basalt or andesite to be generated from magma and, consequently, from ferrimagnetic minerals (Sisson and Grove, 1993). Regarding the low κ values on tuffite, this igneous rock is formed from volcanic ashes and containing large amounts of poorly crystalline minerals (Fabris et al., 1995), which is difficult to form ferrimagnetic minerals. Poggere et al., (2018), also found low magnetic signature on soil formed from tuffites in Brazilian soils with contrasting rock parent material.



The greater κ values on pyritized-andesites (**Fig. 4, table 3**), can be explained by the presence of pyrite and sulfidation process (Passier et al., 2001). As a result, occurs the formation of pyrite in the pyritized-andesites with formation of ferrimagnetic pyrrhotite and magnetite in the rock (Figueiredo, 2000) contributing to the increasing in κ values.

3.3 Ternary gamma-ray and magnetic susceptibility predicted maps, radionuclides content and κ variability at landscape scale

The predicted maps of $^{40}$K, eTh, and eU are demonstrated in figures 5A, 5B and 5C, respectively. In addition, figures 5D and

5E demonstrate the high resolution predicted ternary gamma-ray and magnetic susceptibility maps. Our initial focus lies on describing the interpretations of the three radionuclides in relation to the gamma-ray response associated to lithological-pedological substrates found in the specific landscapes and geomorphic processes.

**Figure 5.** A) Predicted map of $^{40}$K; B) Predicted map of eU; C) Predicted map of eTh; D) and E) 3D landscape perspective and magnetic susceptibility and gamma-ray ternary image, respectively over part of the Keller Peninsula. By integrating gamma-ray spectrometric images with digital elevation models (DEMs) in 3D perspective views, a comprehensive



visualization emerges, revealing intricate connections between gamma-ray responses, terrain morphology attributes and surface processes (pedogeomorphological and periglacial).


The highest eTh contents, corresponding to areas predominantly in green color, occurred on the basaltic-andesites lithology, less mafic rocks rich in plagioclase and higher quartz contents (**Fig. 5D**). In addition, these areas coincide with a flatter relief located on the highest plateaus of the Peninsula, conditions that favor the pedogenesis of deeper soils and higher clay content. Higher clay contents, in turn, favor eTh adsorption on their specific surfaces, contributing to higher eTh readings on these high

plateaus. On the high plateaus the control of eTh grades and distribution are mainly lithological and pedogeomorphological.

The west beach (on the left) also stood out for presenting high levels of eTh on the lithology of undifferentiated sediments (**Fig. 5E**). The explanation lies in the fact that this area is in the lower parts of the landscape and receives meltwater from channels connected to the high plateaus of the landscape, rich in eTh. Once connected to these plateaus, these fluvioglacial waters carry sediments originated by cryoclasty, rich in eTh, which are concentrated on the beaches to the west of the Peninsula.

On the west beaches, the content and distribution of eTh is geomorphological by erosive and depositional processes in a periglacial environment.

The highest eU contents were observed in steep areas with greater slopes, shallow or absent soils, mainly on the basaltic-andesites and andesitic-basalts lithologies (**Fig. 5E**). These geomorphological conditions mean that the eU contents mainly reflect the chemical composition of the rock parent material, where there is a geomorphological and lithological control of the

concentration and distribution of this radionuclide in the landscape. In the steep areas there are erosive and depositional processes of the materials, in such a way that the cryoclasted and eroded materials are deposited on lower plateaus in the landscape. However, this deposition occurs homogeneously on lower altitude plateaus, as there are no preferential flows and material concentrators, unlike fluvioglacial channels that act on the concentration of eTh on the west beach of the Peninsula.

The highest $^{40}$K contents occur over the lower parts of the landscape, mainly on the lower plateaus and beaches in the

southeastern area of the Peninsula, over andesitic-basalts and diorites (**Fig. 5E**). On the other hand, pyritized-andesite lithology presents the lowest levels of $^{40}$K. The east and southwest sides of the Peninsula do not have fluvioglacial drainage channels that concentrate sediments rich in other radionuclides such as eU and eTh. Instead, the east and southeast sides of the Peninsula receive sediments from less mafic rocks (andesitic-basalts and diorites) with higher contents of $^{40}$K, resulting in its concentration. Additionally, natural acid drainage in local fluvioglacial channels, formed by the sulfurization process in the

areas of pyritized-andesites, accelerates the chemical weathering intensity, favoring losses of $^{40}$K content. A recent study undertaken by Mello et al., (2023) found similar results for $^{40}$K content and distribution in sulfate-affected landscapes in Keller Peninsula. In our case, there is lithological and pedogeochemical control over the contents and distribution of $^{40}$K in the area. As demonstrated by Wilford and Minty, (2006) and Dickson and Scott, (1997), ternary gamma imaging associated with DEM can significantly improve the separation and interpretation of radionuclide responses in different lithologies, soils, as well as

periglacial and geomorphological processes that occur in the landscape (Mello et al., 2023b). It mainly focuses on the complex pattern of radionuclides in the landscape associated with these environmental variables. In their publication, Dickson and Scott,





(1997) provided an overview of the distribution of $^{40}$K, $^{232}$Th, and $^{238}$U in parent rocks across Australia. They concluded that the radionuclide content in the rocks explains a significant portion of the variation in gamma signals observed at the land surface. The authors emphasize significant variations within rock classes, indicating that granites do not exhibit a unique

fingerprint. Rawlins et al., (2012) quantitatively reported that rocks accounted for 52% of the variability in gamma radiation observed in the entire Northern Ireland region based on an airborne gamma-ray survey. It was found that felsic rocks generally exhibit higher content of $^{40}$K, eU, and eTh (Rawlins et al., 2007).

The most recent work involving levels and distribution of radionuclides and their relationships with lithology, relief and weathering and pedogenesis processes were carried out by Ribeiro et al., (2018); Souza et al., (2021); Guimarães et al., (2021);

Mello et al., (2020); Mello et al., (2021); Mello et al., (2022); Mello et al., (2022b, 2022a). Despite these works being carried out in a tropical environment, these researchers found similar relationships to explain the distribution and variability of radionuclide contents in the landscape, governed by lithology: when the intensity of pedogenesis weathering processes are little evolved; governed by relief: in areas with predominance of erosion and sediment deposition and, governed by the processes of weathering and pedogenesis: where the soils are more evolved and chemically weathered. However, work carried

out with proximal gamma spectrometry and with this approach practically does not exist in Antarctica.

The κ values in general did not show an apparent pattern on the map and demonstrated high spatial variability between different lithologies, soils and relief features (**Fig. 5D**). However, some areas stood out for presenting higher κ values associated with pyritized-andesites lithologies and andesitic-basalts concomitantly with steep areas and/or those that receive little influence from sediments from other parts of the landscape (**Fig. 5D**). The presence of shallow drift deposits or mixing has likely

disrupted the κ values for numerous rocks in this specific case. It is plausible that the formation of ferrimagnetic materials in periglacial landscapes has been impeded due to reduced Fe release by low weathering intensity (Schwertmann, 1988). Conversely, the higher κ values observed in pyritized-andesites are a consequence of sulfidation (Passier et al., 2001), which leads to the development of pyrite, pyrrhotite, and magnetite in these rocks. Furthermore, increased chemical weathering (Figueiredo, 2000) concentrates ferrimagnetic minerals, consequently contributing to the higher κ values.

The lowest values of κ were observed in areas with Cryosols, which are young soils with little pedogenetic evolution (**Fig. 5D**). The periglacial processes of freezing and thawing of permafrost, along with continuous waterlogging conditions during the thawing phase, hinder the formation of ferrimagnetic minerals. Similarly, Daher et al., (2019) reported low κ values in soils derived from igneous rocks in Antarctica. These values were attributed to the relatively young age of these poorly weathered and pedogenetically evolving soils.


### 3.3 Applicability of geophysical techniques on soil-lithological survey and understanding of periglacial processes

A relationship between radionuclide content/distribution and κ in the landscape in a digital elevation model are demonstrated in figures 6 and 7, respectively). Rock color variations between different lithologies was also observed in the field (**Fig. 8**). The content and distribution of radionuclides and κ are occasionally associated with the lithology of the area, making it difficult

to establish a direct and generalist relationship between radionuclides and κ with the lithological units. This method allows for



the estimation of apparent surface concentrations of naturally occurring radionuclides and their relationship with lithology, pedogeomorphological and periglacial processes (Mello et al., 2023b). By assuming that the absolute and relative concentrations of these radioelements vary significantly with lithology (Dickson and Scott, 1997; Wilford et al., 2016), gamma-ray spectrometric surveys can be used effectively for lithological mapping (Elawadi et al., 2004). However, in this particular

study, the surface lithology is difficult to be map due to multiple geomorphological and periglacial processes that operates in M.A.

Gamma-ray ternary image combined with 3D landscape perspective in different views highlighting areas with higher and lower values of radionuclides (**Fig. 6**). The predicted ternary gamma-ray map (composite image) technique by machine learning was employed to simultaneously display three parameters of radioelement concentrations and distributions on a single image (**Fig.**

**6**). By utilizing color differences, this technique proved effective in discerning periglacial and pedogeomorphological processes associated to lithology and not only lithology (**Fig. 6**). This methodology allowed the identification of areas where distinct surface processes operate where different lithofacies occur within the larger mapped region and detailed studies involving surface process by using gamma-ray spectrometry and κ should be encouraged.






**Figure 6.** Gamma-ray ternary image combined with 3D landscape perspective in different views highlighting areas with higher and lower values of radionuclides. 1: higher eTh content; 2: higher eU content; 3A: higher $^{40}$K content; 3B miner $^{40}$K content in natural sulfate-affected areas (three-dimensional relief generated by ArcGIS).

Regarding κ, surface pedogeomorphological and periglacial processes also influence the distribution of magnetic susceptibility in the landscape, such that the spatial variability of κ has specific relationships with the lithology of the area (**Fig. 7**). Low values may not reflect the properties of the in-situ lithology, as many of the areas are affected by depositional processes caused by periglacial erosion, resulting in the mixing of surface materials (Mello et al., 2023). In a similar vein, Joju et al., (2023) conducted research and discovered that coarse soils in Larsemann Hills, East Antarctica, are primarily composed of magnetic

minerals originating directly from the parent material, showcasing the strong influence of lithology on soil composition. Furthermore, despite the milder and moister climate in the maritime Antarctic region, Lee et al., (2004) observed minimal chemical weathering of bedrocks, suggesting that the soils mainly consist of physically weathered minerals and rock fragments. Moreover, our findings align with those of Warrier et al., (2021a), who argued that while pedogenesis is indeed occurring, its intensity is insufficient to generate magnetic grains.

In some areas, the sulfurization process, induced by the influence of pyritized-andesite (**Fig. 7**), leads to significant environmental acidification and consequential mineralogical transformations affecting κ values (Souza et al., 2012; Lopes et al., 2019). This process may have played a role in the limited occurrence of ferrimagnetic minerals and their uniform distribution across the landscape contributing low variety in κ values (Mello et al., 2023b). Certain regions situated in the lower sections of the terrain are surrounded by mafic igneous rock (andesitic-basalts) in sloping areas, where periglacial erosion rates

are high affect ferrimagnetic minerals distribution over landscape (Francelino et al., 2011; Mello et al., 2023) On the other hand, some areas are located on marine terraces composed of undifferentiated sediments, exhibiting diverse κ values patterns (Mello et al., 2023). The variation in κ values can be attributed to the presence of different sediment types with distinct mineralogical compositions in these specific locations.

    It is also notable the occurrence of low κ values in the elevated and flat parts of the landscape (**Fig. 7**), where Cryosols occur.

The permafrost in this compartment of the landscape hinders ferrimagnetic minerals formation. Water derived from snow melt during summer infiltrates through soil pores and accumulated in the active layer due to low permeability of permafrost. The saturation of soil induces gleyzation and avoid ferrimagnetic minerals precipitation (Zhu et al., 2021). In addition, the presence of a deeper regolith associated with periglacial processes of freezing and thawing of the active layer of permafrost, increases the differences between content and distribution of ferrimagnetic minerals on the surface and ferrimagnetic properties of the

lithology (Mello et al., 2023).

    The sensors were able to detect some lithological transitions, with significant changes in radionuclide and κ contents (**Fig. 8**). However, the sensors do not present values directly associated with lithology due to the high intensity of surface

pedogeomorphological and periglacial processes, it exerts a great influence on geophysical readings in agreement with Dickson and Scott, (1997); Mello et al., (2020) and Mello et al., (2021).


**Figure 7.** Magnetic susceptibility combined with 3D landscape perspective in different views highlighting areas with higher and lower κ values. 1: areas with high κ values; 2 areas with lower κ values over Cryosols (three-dimensional relief generated by ArcGIS).










**Figure 8.** Examples of lithological transitions in Keller Peninsula. A: pyritized-andesite/ basaltic-andesites; B and C: pyritized-andesite/ andesitic-basalts; D and E: pyritized-andesite/ tuffites; F: pyritized-andesite/ andesitic-basalts; G: pyritized-andesite/basaltic-andesites; H: pyritized-andesite/andesitic-basalts; I: pyritized-andesite/diorite; J: undifferentiated marine sediments; L: tuffite/ pyritized-andesite; M: andesitic-basalts/ basaltic-andesites; N: undifferentiated marine sediments/pyritized-andesite; O: pyritized-andesite/ basaltic-andesites; P: andesitic-basalts/ pyritized-andesite/ undifferentiated marine sediments.

### 3.4 Study limitations and recommendations

Figure 9 demonstrates the coefficient of variation (prediction error) of the ternary gamma-ray and κ maps. The relatively low coefficient of variation values in our study can be attributed to the nested-LOOCV technique. These maps, associated with the CCC (table 2), illustrates the limitations of the models in predicting and spatializing geophysical data. The prediction errors were low for the geophysical variables, in agreement with the high CCC values shown in table 2, however, such errors do exist. It is possible to observe that the main prediction errors are associated with the steepest areas of the Peninsula, while the smallest are associated with areas with smoother to flat slopes. This shows that the main limitation of the modeling is related to the small number and distribution of samples read with the geophysical sensors. In this context, the relatively limited sample number as well as the distribution of samples is justified by the adverse field conditions (e.g., steep areas with snowbanks, glaciers, sharp rocks and frozen ground combined with high slopes, resulting in high danger areas for data acquisition by using proximal sensors). In other words, the logistical difficulties imposed by cold environments in field conditions were one of the significant limitations of this work, as noted by Fisher, (2014; 2015) and Mello et al., (2023). However, moderate CCC values and satisfactory CV in modeling processes, an exploratory evaluation for field data acquisition can provided informative results (Dharumarajan et al., 2017; Khaledian and Miller, 2020; Mansuy et al., 2014; Mosleh et al., 2016; Poggio et al., 2016).





**Figure 9:** a) Coefficient of variation for magnetic susceptibility predicted map; b) Coefficient of variation for ternary gamma-ray predicted maps.

The low number of samples in this study (87) was not so appropriate for a more specific approach. However, the RF algorithm combined with nested-LOOCV were appropriate for small samples number, as demonstrated in other researches (Mello et al., 2022a; Mello et al., 2022b, 2022c). In addition, in-situ evaluation brings several uncontrolled factors (such as rocks or





fragments mixing due to periglacial erosion, permafrost activity, fluvioglacial channels and others), can impact the prediction and reduce the CCC and increase CV (Mello et al., 2023b).

Another limitation of this study is the unavailability of spatially continuous detailed lithological map, which affects the prediction performance (CCC, table 2) and CV maps (**Fig. 9**). Furthermore, the variability of sensor readings is another limitation, which is little, but it exists. As a result, this variability can reduce the accuracy of the information. Nevertheless,

our methodology tackled this concern by extending the reading time of the gamma-ray sensor to 3 minutes and employing the mean values of three magnetic susceptibility readings. Mello et al., (2023), carried out a similar approach where the same errors and experimental conditions were observed when modeling the intensity of weathering and studying pedogenesis in soil profiles in Keller Peninsula, using machine learning algorithms. These researchers also adjusted the data collection method with the same geophysical sensors used in this research.

The applicability of the findings here, however, is restricted to comparable environments, specifically those exhibiting periglacial conditions, igneous lithology, similar precipitation, temperature, and relief patterns. Given that many of the Maritime Antarctica Islands and some parts of Antarctic Peninsula share these common or similar environmental features, it is strongly recommended to promote similar geophysical survey characterization efforts.

**4. Conclusion**

The research introduced a structured approach to specialize geophysical variables using machine learning techniques. It has been demonstrated that employing machine learning methodologies is promising for accurately mapping natural gamma-ray radioactivity and magnetic susceptibility characteristics. Through our methodology, we fitted regression models that identified key predictors, assessing accuracy and uncertainty across the RF model and ensuring consistent predictions through multiple

pedogeoenvironmental iterations.

The RF algorithm was efficient and successfully predicted detailed maps of gamma-spectrometric and magnetic susceptibility variables in periglacial environments with diverse igneous rock substrates. Relief-related morphometric variables significantly influenced the distribution of radionuclides and ferrimagnetic minerals on the land surface. The nested-LOOCV method proved suitable for geophysical data with limited samples, providing robust evaluation of algorithm performance and generating

accurate and high-performing mean maps.

The highest levels of eTh were observed in three key areas: the elevated parts of the landscape, the flat areas, and the west beach. The west beach receives detrital materials from periglacial erosion, which come through fluvioglacial melting channels from the eTh-rich elevated parts. The eTh contents are controlled by lithology and pedogeomorphological processes.

The highest eU contents were observed in the steepest areas, characterized by the greatest slope, forming a ring around the

highest parts of the landscape. In this case, the control of eU contents is determined by lithology and geomorphological processes, such as rock cryoclasty, periglacial erosion, and heterogeneous Accumulation of materials in the lower elevations of the terrain.



The highest levels of [40]K were found in the most felsic rocks and areas with minimal influence from material deposition caused by periglacial erosion. Conversely, the lowest contents of [40]K were observed in regions affected by the pedogeochemical process of sulfurization, specifically on pyritized-andesite within/around fluvioglacial melting channels. The control of [40]K levels is determined by both lithology and pedogeochemical processes.

The $\kappa$ did not exhibit an apparent distribution pattern, although the highest levels were observed in pyritized-andesites areas, while the lowest levels were found in Cryosol areas. Pyritized-andesite facilitates the release of iron in the system through sulfurization and contains associated pyrrhotite, which contributed to higher $\kappa$ values. On the other hand, Cryosols, in addition to increasing the distance between surface materials and the rocky substrate, experience seasonal freezing and thawing activity of the active permafrost layer, creating conditions that discourage the formation of ferrimagnetic minerals and reduce $\kappa$ values. The control of $\kappa$ values is determined by lithology and pedological-periglacial processes associated with Cryosols.

In areas with diverse terrain attributes and a prevalence of active and intense periglacial processes, the predicted-spatialized geophysical variables do not accurately represent the lithological composition of the substrate. This is because the various periglacial processes in the region, combined with the morphometric characteristics of the landscape, work to redistribute, mix, and homogenize the surface materials.

## 5. Acknowledgements

We express our gratitude to the National Council for Scientific and Technological Development (CNPq) for providing the first author's scholarship and essential resources through grant No. 134608/2015-1 and grant number 305996/2018-5. Additionally, we acknowledge partial funding for this study from the Coordenação de Aperfeiçoamento de Pessoal de Nível Superior - Brazil (CAPES) - Finance Code 001.

We extend our thanks to the Terrantar – UFV group, Geotechnologies in Soil Science group, Laboratório de Geoprocessamento e Pedometria – UFV (LabGeo – UFV), 'Programa de Pós-Graduação em Solos e Nutrição de Plantas – PGSNP' of the Soil Department of the Universidade Federal de Viçosa, Brazil, and Programa Antártico Brasileiro for their valuable support.

## 6. Declaration of Generative AI and AI-assisted technologies in the writing process

Statement: During the preparation of this work, the authors utilized GPT-4 to correct the grammar and structure of the English language, ensuring clarity in the sentences for the reader. After using this tool/service, the author reviewed and edited the content as needed and take full responsibility for the content of the publication.

## 7. Code and data availability

All analyses and codes used in this research were developed in R software version 4.0.3 (R Core Team, 2023; Kuhn et al., 2013). The codes and data used in this research can be found at https://zenodo.org/doi/10.5281/zenodo.10828281 (Moquedace et al., 2024). All packages used in the R software, as well as their respective versions, are listed in the database, and codes are available in the data_base.zip file in the indicated repository.





## 8. Authors contribution

**Danilo César de Mello:** conceived of the presented idea, carried out the experiment, developed the theoretical formalism, contributed to the design and implementation of the research, to the analysis of the results and to the writing of the manuscript. He provided critical feedback and helped shape the research, analysis and manuscript.

**Clara Glória Baldi:** designed the model and the computational framework and analysed the data, planned and carried out the simulations, performed the analytic calculations and performed the numerical simulations, modelling processing, evaluate algorithms performance, variables importance and statistical analyses.

**Cassio Marques Moquedace:** contributed to the interpretation of the results, took the lead in writing the manuscript. Devised the project, the main conceptual ideas and proof outline. He worked out almost all of the technical details. All authors provided critical feedback and helped shape the research, analysis and manuscript.

**Isabele de Angeli Oliveira:** contributed to the interpretation of the results, took the lead in writing the manuscript. All authors provided critical feedback and helped shape the research, analysis and manuscript.

**Gustavo Vieira Veloso:** contributed to the interpretation of the results, took the lead in writing the manuscript. All authors provided critical feedback and helped shape the research, analysis and manuscript.

**Lucas Carvalho Gomes:** performed the analysis, drafted the manuscript and designed the figure. All authors provided critical feedback and helped shape the research, analysis and manuscript.

**Márcio Rocha Francelino:** Provided de financial support, leadership of the group, critical revision of the article. He contributed to the interpretation of the results and verified the analytical methods. Encouraged the co-authors to investigate a specific aspect and supervised the findings of this work.

**Carlos Ernesto Gonçalves Reynaud Schaefer:** Critical revision of the article. All authors discussed the results and commented on the manuscript. He contributed to the interpretation of the results and verified the analytical methods.

**Elpídio Inácio Fernandes Filho:** Critical revision of the article. He designed the model and the computational framework and analysed the data. He contributed to the interpretation of the results and verified the analytical methods. All authors discussed the results and commented on the manuscript.





**José João Leal Lelis de Souza:** Critical revision of the article. He contributed to the interpretation of the results and verified the analytical methods. All authors discussed the results and commented on the manuscript.

**Edgar Batista de Medeiros Júnior:** Critical revision of the article. He contributed to the interpretation of the results and verified the analytical methods. All authors discussed the results and commented on the manuscript.

**Fabio Soares de Oliveira:** Critical revision of the article. He contributed to the interpretation of the results and verified the analytical methods. All authors discussed the results and commented on the manuscript.

**Tiago Osório Ferreira:** Critical revision of the article. He contributed to the interpretation of the results and verified the analytical methods. All authors discussed the results and commented on the manuscript.

**José Alexandre Melo Demattê:** Critical revision of the article. He contributed to the interpretation of the results and verified the analytical methods. All authors discussed the results and commented on the manuscript.

### 9. Competing interests

The authors have the following competing interests: At least one of the (co-)authors is a member of the editorial board of Geoscientific Model Development.

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
