# Peer review of "Proximal Surface Pedogeophysical Characterization in Maritime Antarctica: Assessing Pedogeomorphological, Periglacial, and Landform Influences"

_Geoscientific Model Development, 2024_

## Referee Comment (RC2)

**Application of machine learning to proximal gamma-ray and magnetic susceptibility surveys in the Maritime Antarctic: assessing the influence of periglacial processes and landforms**

Danilo C. de Mello1; Clara G. O. Baldi1; Cássio M. Moquedace1; Isabelle de A. Oliveira1; Gustavo V.
Veloso1; Lucas C. Gomes3; Márcio R. Francelino1; Carlos E. G. R. Schaefer1; Elpídio I. Fernandes-Filho1; Edgar B. de Medeiros Júnior1; Fabio S. de Oliveira4; José J. L. L. de Souza1; Tiago O. Ferreira2; 
[revised manuscript text omitted]

---

## Author Comment (AC1)

**RC2**: 'Comment on gmd-2024-2', Anonymous Referee #2, 24 Jun 2025

**This study presents a relevant contribution to the modeling of geoscientific data using machine learning, integrating information from both literature sources and field-based geophysical surveys. Its significance is further enhanced by its focus on a scientifically important and logistically challenging region. In this context, the application of data modeling techniques is particularly valuable. The study also incorporates a robust model validation approach.**

**Answer**: *We appreciate the reviewer's contribution to improving the quality of our manuscript. Indeed, it does not make sense to list all variables if only 20 were used in the modeling process. Accordingly, we have revised the table to include only the variables actually used, as recommended.*

**However, the manuscript requires major revision. The research objective should be carefully reconsidered — the production of a map is not an end in itself, but rather a tool to support the spatial analysis and understanding of underlying geoscientific processes. Moreover, the text contains redundant, out-of-context, and even repeated phrases and paragraphs, which compromise clarity and coherence.**

**Answer:** *We agree with the reviewer and review the entire objectives section, considering the considerations highlighted. follows the revised version of the objectives:*

*"Given the above, this study aimed to enhance the geoscientific understanding of periglacial and pedogeomorphological processes in Keller Peninsula (Maritime Antarctica) by integrating radiometric and magnetic surficial data with advanced spatial analysis techniques. Specifically, we sought to: i) investigate how natural radioactivity (238U, 232Th, and 40K) and magnetic susceptibility (κ) vary across heterogeneous lithological and pedological substrates and how they reflect underlying geomorphic and periglacial dynamics; ii) evaluate the performance of a machine learning algorithm, combined with the nested Leave-One-Out Cross-Validation method, in modeling and spatializing geophysical surface data; iii) use the generated gamma-ray ternary and κ maps as tools to interpret and reveal spatial patterns related to soil development, rock weathering, and cryogenic processes in the study area."*

*Furthermore, the entire manuscript has been carefully revised in an effort to address the issues raised by the reviewer regarding redundancy, lack of context, and repeated phrases or paragraphs, which compromised the clarity and coherence of the text.*

*We carefully reviewed the entire manuscript to address the issues of redundancy, lack of contextual clarity, and repeated content. Several phrases and paragraphs were rewritten, removed, or reorganized to improve clarity, coherence, and overall readability. For future revisions, we would greatly appreciate it if you could kindly indicate the specific sentences or paragraphs, you believe should be removed or modified. This will help us address your concerns more precisely and efficiently. We believe the current version of the manuscript has been substantially improved based on your comments. If necessary, we are also willing to submit the manuscript to a professional proofreading service specialized in geosciences to ensure the highest linguistic and editorial quality.*

**The results indicate a lack of correlation between lithology and the geophysical variables (K, k, eU, and eTh), while strong correlations are observed with terrain attributes. This outcome warrants in-depth discussion, as in a setting where the pedon is poorly developed, one would typically expect a stronger correlation between primary mineralogy and the concentrations of radionuclides or magnetic minerals. The absence of robust mineralogical, petrographic, and geochemical data or citations significantly weakens the technical discussion and, to some extent, limits the depth of interpretation of the proposed model. There is also a need to enhance the comparison between the existing geological map of the study area and the field sampling points, in order to clarify the different compartments identified in the ternary radionuclide map.**

*Answer: We agree with the reviewer's observation. Indeed, in environments where soils are weakly developed pedogenetically, as is the case in much of our study area, which is characterized by low chemical weathering intensity in most of the analyzed profiles, a strong correlation between soil radionuclide content and the parent material is expected, with lithology acting as the main controlling factor. However, in regions within the study area that are dominated by rugged terrain and steep slopes, the influence of intense geomorphic and pedogeomorphological processes, such as active periglacial erosion, freeze–thaw cycles, fluvioglacial drainage, and cryoturbation makes topography a more decisive factor than lithology in the redistribution of materials and radionuclides across the landscape. As a result, in certain portions of Keller Peninsula, the observed*

*radionuclide concentrations do not match the values that would be expected based on the lithological types described in the literature. Similarly, the magnetic susceptibility values do not always directly reflect the parent material, as the aforementioned surface processes along with sulfidation interfere with the formation, transformation, and redistribution of ferrimagnetic minerals in the environment.*

*We acknowledge that the reviewer is correct, and we agree that the manuscript would benefit from greater emphasis in the methodology, results and discussion, limitations, and conclusions sections to clarify details regarding the lithological information and mapping used in our study, along with a more in-depth discussion. In response, we have added the following information to the revised version:*

**Methodology**

*"We used the detailed geological-lithological map of Admiralty Bay and its surrounding area at a 1:50,000 scale, produced by British geologists between 1948 and 1960 (Birkenmajer, 1980), as the base map for lithological interpretation. To enhance the analysis, mineralogical, petrographic, and geochemical information on the various rock units of Keller Peninsula such as basaltic-andesites, andesitic-basalts, tuffites, diabase, pyritized-andesites, and diorites rocks was extracted from Birkenmajer (1980). These data supported the interpretation of bedrock as a key soil-forming factor, particularly in the mobilization and spatial distribution of radionuclides ($^{238}$U, $^{232}$Th, and $^{40}$K) in surface soils. The mineralogical composition of the parent material, especially the presence of Fe-bearing silicates and accessory minerals, influenced the geochemical dynamics of secondary ferrimagnetic mineral formation and the inheritance of primary minerals. The spatial relationship between sampling points and lithological boundaries was assessed using a detailed lithological map, revealing distinct geological compartments corresponding to variations in landscape relief.*

*The lower portions of the landscape consist mainly of marine terraces. At intermediate elevations, lithified pyroclastic deposits, known as tuffites, are predominant. These tuffites are characterized by volcanic glass shards, plagioclase, and pyroxene crystals, as well as lithic clasts embedded in a fine ash matrix. They frequently exhibit varying degrees of alteration, including chloritization and sericitization, and may be cemented by secondary silica or calcite (Nawrocki et al., 2021).*

*Above the tuffites, extensive outcrops of andesitic-basalts and basaltic-andesites dominate the upper landscape. These volcanic rocks primarily consist of labradorite-andesine phenocrysts set within a groundmass of plagioclase, volcanic glass, and clinopyroxene (Nawrocki et al., 2021). Scattered throughout these units are occurrences of pyritized andesites, which have undergone significant post-magmatic hydrothermal alteration. This alteration transformed primary plagioclase and pyroxene into secondary minerals such as chlorite, albite, carbonate, and quartz. Additionally, quartz–pyrite mineralization developed within these andesites (Birkenmajer, 1980).*

*Less abundant and restricted to specific localized zones, diorite outcrops occur notably on Keller Peninsula. These diorites are composed mainly of plagioclase (andesine to labradorite), hornblende, and minor biotite, with accessory minerals such as magnetite, apatite, titanite, and zircon. The texture is generally coarse-grained and equigranular (Birkenmajer, 1980; Valeriano et al., 2008)."*

***Results and Discussion (end of Section 3.1) Model's performance and variable's importance):***

*"Our results regarding the distribution of radionuclides across the landscape differ slightly from those commonly reported in the literature (Dickson and Scott, 1997; Mello et al., 2021; Wilford and Minty, 2006; Wilford and Thomas, 2012), which typically reports a strong correlation between radionuclide concentrations and the parent material in poorly developed soils. Although the low chemical weathering intensity observed in our study area suggests that lithology should exert primary control, the presence of highly dissected terrain, steep slopes, and active periglacial processes including periglacial erosion, freezing-thawing cycles, and cryoturbation intensifies the influence of topography on the redistribution of radionuclides. As a result, in certain areas of Keller Peninsula, radionuclide concentrations in soils deviate from the expected values based solely on the underlying rock types. Practically all the relief variables are associated with the landform that control the surface periglacial and pedogeomorphological processes of the Keller Peninsula landscape. Periglacial erosion, glacial fluvial melt channels, freezing and thawing of the active layer of permafrost and solifluxion are the most frequent periglacial and pedogeomorphological processes in Keller Peninsula, as observed by Francelino et al., (2011) and López-Martínez et al., (2012). These processes promote the fragmentation, redistribution and mixing of materials in significant areas of the landscape (Mello et al., 2023; Mello et al., 2023), which can contribute to variations*

*in radionuclide and k values, as well as increase prediction errors in the points of greater occurrence of these processes, such as the sloping areas of the landscape (Mello et al., 2022). The same periglacial processes and landscape dynamics also influence iron geochemistry. As a result, soils and areas underlain by mafic rocks such as basaltic-andesite and andesitic-basalt may exhibit relatively low concentrations of ferrimagnetic minerals, which is reflected in lower magnetic susceptibility readings (Fig. 4). The opposite can also occur; for example, soils developed over pyritized andesite show higher magnetic susceptibility values, indicating greater concentrations of ferrimagnetic minerals (Fig. 4).*

**Results and Discussion: section 3.2 Radionuclides and $\kappa$ contents on lithological compartments and their relationship with mineralogy:**

*"The spatial patterns of natural radioactivity and magnetic susceptibility across Keller Peninsula are more strongly influenced by topography than by lithology. In steep, periglacially active terrains, geomorphic and pedogeomorphological processes such as cryoturbation, freeze–thaw cycles, and periglacial erosion promote the downslope transport and mixing of soil and minerals, resulting in the redistribution of radionuclides and ferrimagnetic minerals independent of bedrock type. Birkenmajer's (1980) geological mapping and petrographic studies further support that variations in mineral assemblages, especially between lightly altered mafic rocks and hydrothermal zones, and the presence of secondary minerals such as zeolites, albite, and iron oxides contribute to these patterns. Consequently, geophysical signals (e.g., eU, eTh, $^{40}K$, and magnetic susceptibility) often reflect a mixed mineralogical signature redistributed by topographic and geomorphological dynamics, rather than a direct inheritance from the parent material. This may explain our observations such as unexpectedly low magnetic susceptibility over mafic rocks and elevated values over altered andesites, underscoring the dominant role of relief and periglacial processes in shaping geophysical variability in Keller Peninsula."*

*Furthermore, the entire Section 3.3, Ternary Gamma-Ray and Magnetic Susceptibility Predicted Maps, Radionuclide Content, and $\kappa$ Variability at Landscape Scale, has been thoroughly rewritten to clarify the predominant influence of surface processes over lithology, directly addressing the reviewer's insightful comment.*

*"The highest eTh values, predominantly represented by the green areas on the map, occur mainly over basaltic-andesite lithologies, rocks that are less mafic and richer in plagioclase and quartz (Fig. 5D). These regions coincide with flatter, high-elevation plateaus where deeper soils with higher clay content develop. The increased clay fraction enhances the adsorption capacity for eTh onto soil particle surfaces, thereby elevating eTh readings in these high plateau zones. In such areas, the spatial distribution and concentration of eTh are primarily controlled by lithology and pedogeomorphological factors.*

*In contrast, the western beach area, located at lower landscape positions, also exhibits elevated eTh levels associated with undifferentiated sediments (Fig. 5E). This pattern is explained by the geomorphological setting where fluvioglacial meltwaters originating from the high plateaus transport cryoclastically derived, eTh-rich sediments downslope. These sediments accumulate on the western beaches, demonstrating how erosive and depositional processes in a periglacial environment govern the distribution of eTh in this sector.*

*Regarding eU, the highest values are found on steep slopes characterized by shallow or absent soils, mainly over basaltic-andesites and andesitic-basalts lithologies (Fig. 5E). In these geomorphologically active areas, eU distribution largely reflects the chemical composition of the bedrock, indicating strong lithological and geomorphological control. Cryoclastically fractured and eroded materials are transported downslope by periglacial erosion and deposited more homogeneously across lower plateaus, differing from the focused sediment transport through fluvioglacial channels responsible for eTh concentration on the west beach.*

*The $^{40}K$ values peaks predominantly in lower landscape positions, including lower plateaus and southeastern beaches, where andesitic-basalts and dioritic lithologies prevail (Fig. 5E). Conversely, pyritized-andesite zones show the lowest $^{40}K$ values, likely due to enhanced chemical weathering driven by natural acid drainage and sulfurization processes in local fluvioglacial channels. These processes accelerate potassium depletion, as observed in recent studies of sulfate-affected landscapes in Keller Peninsula (Mello et al., 2023). Therefore, both lithological composition and pedogeochemical processes regulate $^{40}K$ distribution in the area.*

*Previous research (Wilford and Minty, 2006; Dickson and Scott, 1997) has demonstrated that combining ternary gamma imaging with digital elevation models improves the interpretation of radionuclide spatial patterns by integrating lithological, soil, periglacial, and geomorphological influences (Mello et al., 2023b). Dickson and Scott (1997) showed that rock radioelement contents explain much of the gamma radiation variability, while also highlighting intra-class heterogeneity—granites, for example, lack a unique radionuclide signature. Similarly, Rawlins et al. (2012) quantified that bedrock type accounted for 52% of gamma radiation variability across Northern Ireland. Felsic rocks generally exhibit elevated eU, eTh, and $^{40}$K contents (Rawlins et al., 2007). Recent tropical environment studies (Ribeiro et al., 2018; Souza et al., 2021; Guimarães et al., 2021; Mello et al., 2020, 2021, 2022a,b) have linked radionuclide variability to lithology in areas with minimal pedogenetic alteration, to relief in erosion and sediment deposition zones, and to weathering and pedogenesis in well-developed soils. However, recent studies, including the first applications of gamma spectrometry and magnetic susceptibility to Antarctic soils undertaken by (Mello et al., 2023; Mello et al., 2023), have suggested a strong influence of topography on the distribution of geophysical variables, which were thoroughly detailed in this work.*

*Magnetic susceptibility (κ) values exhibit high spatial variability across lithologies, soils, and landforms, showing no consistent broad-scale pattern (Fig. 5D). Nonetheless, localized zones of elevated κ correlate with pyritized-andesite and andesitic-basalt lithologies, particularly on steep slopes or areas minimally influenced by sediment influx from other parts of the landscape. The widespread presence of shallow drift deposits and sediment mixing likely disrupts κ patterns across many lithologies. In periglacial settings, limited chemical weathering and reduced iron release hinder ferrimagnetic mineral formation (Schwertmann, 1988). Conversely, sulfidation in pyritized andesites promotes the development of pyrite, pyrrhotite, and magnetite, enhancing magnetic susceptibility (κ) (Passier et al., 2001). This effect is further intensified by higher iron availability and chemical weathering, which together concentrate ferrimagnetic minerals and contribute to increased susceptibility values (Figueiredo, 2000; Mello et al., 2023).*

*The lowest κ values occur in areas dominated by Cryosols, young soils with minimal pedogenetic development (Fig. 5D). Freeze-thaw cycles of permafrost combined with prolonged waterlogging during thaw phases inhibit ferrimagnetic mineral formation. This pattern aligns with findings by Daher et al. (2019), who reported low κ values in*

*Antarctic soils derived from igneous rocks, attributed to their relatively young age and limited weathering.*

*The spatial distribution of radionuclides and magnetic susceptibility in Keller Peninsula (Fig. 5) results from a dynamic interplay between mineralogical characteristics of the bedrock, topographic controls on soil development and sediment transport, and active periglacial geomorphological processes. These factors collectively modulate the geophysical signatures observed, producing patterns that cannot be solely attributed to lithology but rather to its modification through landscape evolution and pedogeochemical cycling.*

***Results and Discussion: 3.4 Study limitations and recommendations:*** *We have also addressed part of the issue raised by the reviewer in the "Limitations" section, acknowledging it as a limitation of the study. However, we also emphasized and explained how this limitation was properly addressed within our methodological approach.*

*"The absence of detailed mineralogical, petrographic, and geochemical analyses constitutes a limitation of this study. This constraint was primarily due to logistical and operational challenges associated with fieldwork in remote and climatically extreme environments, which limited both the time available for sample collection and the transport of materials for laboratory analysis. Additionally, the main focus of the study was the application and evaluation of predictive models based on surface geophysical data, rather than a comprehensive mineralogical-petrographic characterization. Nevertheless, we mitigated this limitation by incorporating and referencing existing detailed geological studies of the area, which provided essential information on the lithological framework, mineralogy and post-magmatic alteration processes. This information contributed significantly to understanding lithology as both a source of radionuclides and a provider of iron, which plays a key role in the formation of ferrimagnetic minerals either through pedogenetic processes (in the clay fraction) or as an inherited feature from the parent material (in the sand fraction).We recommend that future studies integrate in situ mineralogical and geochemical analyses to deepen the interpretation of the geophysical signals and refine model accuracy.*

***Conclusion:***

*"Although the low degree of pedogenetic development and limited chemical weathering in the study area would typically suggest a strong lithological control over radionuclide*

*concentrations, our findings indicate that topographic factors play a more dominant role. The highly dissected relief, steep slopes, and active periglacial processes, such as erosion and cryoturbation, contribute significantly to the redistribution of materials and radionuclides. As a result, in certain areas of Keller Peninsula, radionuclide concentrations do not align with the expected values based solely on the underlying lithology."*

**2. Methodology**

**2.1 Study Area, Lithological–Soil Surveys and Sampling Points**

**From a methodological standpoint, the review of previous data sources is necessary, particularly regarding the geological map. It is unclear whether the lithological units shown in Figure 2A originate from Birkenmajer (1980) or from a more recent geological survey. If the map in Figure 2A was produced as part of this study, it is essential to explicitly describe the mapping methodology, especially given the lack of petrographic, geochemical, and mineralogical data. For instance, the distinction between basaltic andesite and andesitic basalt typically requires geochemical discrimination diagrams. Such data should be presented or, at the very least, the data source should be properly cited.**

**Answer:** *We agree with the reviewer and have made an effort to address the issue raised. In the revised version, we clarified the concerns pointed out by the reviewer within the Methodology section, as follows:*

*"We used the detailed geological-lithological map of Admiralty Bay and its surrounding area at a 1:50,000 scale, produced by British geologists between 1948 and 1960 (Birkenmajer, 1980), as the base map for lithological interpretation. To enhance the analysis, mineralogical, petrographic, and geochemical information on the various rock units of Keller Peninsula such as basaltic-andesites, andesitic-basalts, tuffites, diabase, pyritized-andesites, and diorites rocks was extracted from Birkenmajer (1980). These data supported the interpretation of bedrock as a key soil-forming factor, particularly in the mobilization and spatial distribution of radionuclides ($^{238}U$, $^{232}Th$, and $^{40}K$) in surface soils. The mineralogical composition of the parent material, especially the presence of Fe-bearing silicates and accessory minerals, influenced the geochemical dynamics of*

*secondary ferrimagnetic mineral formation and the inheritance of primary minerals. The spatial relationship between sampling points and lithological boundaries was assessed using a detailed lithological map, revealing distinct geological compartments corresponding to variations in landscape relief.*

*The lower portions of the landscape consist mainly of marine terraces. At intermediate elevations, lithified pyroclastic deposits, known as tuffites, are predominant. These tuffites are characterized by volcanic glass shards, plagioclase, and pyroxene crystals, as well as lithic clasts embedded in a fine ash matrix. They frequently exhibit varying degrees of alteration, including chloritization and sericitization, and may be cemented by secondary silica or calcite (Nawrocki et al., 2021).*

*Above the tuffites, extensive outcrops of andesitic-basalts and basaltic-andesites dominate the upper landscape. These volcanic rocks primarily consist of labradorite-andesine phenocrysts set within a groundmass of plagioclase, volcanic glass, and clinopyroxene (Nawrocki et al., 2021). Scattered throughout these units are occurrences of pyritized andesites, which have undergone significant post-magmatic hydrothermal alteration. This alteration transformed primary plagioclase and pyroxene into secondary minerals such as chlorite, albite, carbonate, and quartz. Additionally, quartz–pyrite mineralization developed within these andesites (Birkenmajer, 1980).*

*Less abundant and restricted to specific localized zones, diorite outcrops occur notably on Keller Peninsula. These diorites are composed mainly of plagioclase (andesine to labradorite), hornblende, and minor biotite, with accessory minerals such as magnetite, apatite, titanite, and zircon. The texture is generally coarse-grained and equigranular (Birkenmajer, 1980; Valeriano et al., 2008)."*

**3. Results and Discussion**

**3.1 Model Performance and Variable Importance**

**The model identifies 16 terrain attributes as the main predictors for eU, eTh, K, and k concentrations, while lithology did not contribute significantly. Although the model exhibits reasonable performance (except for K), the lack of correlation between lithology and radiation data is a critical point in terms of model interpretation. The authors themselves cite Gregory and Horwood (1961), stating**

that "the mineralogy and geochemistry of the substrate present the greatest contribution to radionuclide contents".

In this regard, given the input dataset — comprising 48 terrain attributes (~8 mm² spatial resolution), 3 radiogenic elements (~55 m² per point), 1 magnetic susceptibility attribute (~57 m² per point), and 1 soil variable (20 profiles) — it is necessary to evaluate whether terrain attributes have been overemphasized in the modeling process.

This concern is reflected in Figure 5, where the resulting maps show a strong topographic imprint, resembling a fused radiometric–topographic product, which is not typical of radiation or magnetic susceptibility maps, whose signals are usually controlled by geological units and/or recent sedimentary deposits (e.g., terraces and plateaus).

*Answer: We appreciate the reviewer's critical observation and fully acknowledge the importance of substrate mineralogy and geochemistry in determining natural radionuclide concentrations, as highlighted by Gregory and Horwood (1961). However, the lack of a significant correlation between lithology and the geophysical variables in our model may be attributed to geomorphic, pedogeomorphological and periglacial processes that alter or obscure the surface geophysical signature of the parent material. These land surface processes can significantly modify the original lithological signal in the uppermost soil surface horizons, areas of deposition of colluvial and fluvioglacial materials and surface cryoclastic materials. These surface processes may partially explain the weak contribution of lithology to the predictive models, despite its well-established geochemical importance.*

*Furthermore, the inclusion of 48 terrain attributes was guided by previous studies demonstrating the strong control exerted by topography on the spatial distribution of geophysical properties in regions with poorly developed soils. The morphometric covariates derived from a high-resolution digital elevation model (approximately 8 mm² per pixel) were selected to capture microtopographic variations that influence erosional and depositional processes, soil redistribution, and horizon development, all of which directly affect the spatial variability of radionuclides and ferrimagnetic minerals.*

*We nevertheless recognize that this emphasis on terrain attributes may have outweighed the potential influence of lithological factors in the modeling process. For this reason, we*

*have discussed this as a limitation of the study and emphasized the need for more detailed geological inputs in future research such as higher-resolution lithological maps, reflectance spectroscopy, or direct lithogeochemical data to better integrate mineralogical controls into predictive frameworks.*

**In this regard, given the input dataset — comprising 48 terrain attributes (~8 mm² spatial resolution), 3 radiogenic elements (~55 m² per point), 1 magnetic susceptibility attribute (~57 m² per point), and 1 soil variable (20 profiles) — it is necessary to evaluate whether terrain attributes have been overemphasized in the modeling process.**

**Answer:** *We respectfully argue that terrain attributes were not overemphasized in the modeling process. Rather, their prominence among the predictors emerged organically from the model's internal variable importance metrics, which assess each covariate's marginal contribution to predictive performance. All environmental variables, including lithology were provided to the model with equal weight and processed under the same cross-validation framework. The high ranking of morphometric variables reflects their superior predictive capacity in this specific geomorphological context, rather than a modeling bias toward topographic data.*

*Additionally, we accounted for differences in spatial resolution and representativeness of each input layer. While terrain attributes were derived from a high-resolution digital elevation model (~20cm), the lithological dataset, extracted from a 1:50,000 geological map, lacked detailed spatial variability at the sub-pedon scale. Therefore, it is likely that the lower performance of lithological variables is due to their coarser and more generalized representation, rather than model misweighting.*

*Moreover, in cold-climate and periglacial environments like our study area, terrain-driven processes (e.g., cryoturbation, slope-driven redistribution, differential weathering) play a disproportionately large role in shaping the surface distribution of radionuclides and ferrimagnetic minerals. Thus, the outcome of our modeling approach aligns with the known environmental dynamics of the region, reinforcing the robustness and ecological plausibility of our results.*

**3.2 Radionuclides and κ Contents in Lithological Compartments and Their Relationship with Mineralogy**

**Although the study presents some correlation patterns, there is no actual comparison with mineralogical data, either generated in the study or sourced from previous work. This omission prevents a more robust interpretation of the relationship between radionuclide distribution and the underlying lithological/mineralogical context.**

**Answer:** *We sincerely appreciate the reviewer's insightful comment. We fully recognize that the absence of detailed mineralogical, petrographic, and geochemical analyses represents a limitation of our study. This constraint primarily stemmed from the logistical and operational challenges inherent to conducting fieldwork in remote and climatically extreme regions such as Maritime Antarctica. These conditions restricted the duration of field campaigns and limited our capacity to collect and transport samples for laboratory-based mineralogical characterization. However, it is important to note that the central focus of this research was not mineralogical or petrographic analysis per se, but rather the application and evaluation of predictive models based on surface geophysical data, specifically, radiometric and magnetic susceptibility measurements in order to understand how these properties reflect the underlying geomorphic, pedological, and periglacial dynamics in Antarctic environment.*

*To mitigate this limitation, we incorporated and carefully referenced existing high-quality geological surveys and peer-reviewed studies, which provide detailed lithological descriptions, mineralogical compositions, and insights into post-magmatic alteration processes for the study area. This existing body of knowledge was instrumental in supporting our interpretations, particularly in understanding lithology as both a source of natural radionuclides and a key contributor of iron, which influences the formation of ferrimagnetic minerals either inherited from the parent rock (notably in the sand fraction) or formed secondarily through pedogenetic processes (notably in the clay fraction).*

*It is important to highlight that this limitation has been explicitly acknowledged in the revised manuscript (Limitations section), along with a discussion of its implications for the interpretation of our findings. We also emphasize that the integration of in situ mineralogical, petrographic, and geochemical data represents a valuable direction for*

*future research, as it would allow for a more refined interpretation of the geophysical signals and contribute to enhancing the robustness and reliability of the proposed model.*

---

## Author Comment (AC2)

**RC1**: 'Comment on gmd-2024-2', Anonymous Referee #1, 04 Jan 2025

**Dear Editor,**

**The topic of the article is suitable for the Journal.**

**The technical aspect is good.**

**Language and structure is good.**

**Statistical design is correct.**

**Discussion is very good.**

**Answer**: *We appreciate the reviewer's contribution to improving the quality of our manuscript. Indeed, it does not make sense to list all variables if only 20 were used in the modeling process. Accordingly, we have revised the table to include only the variables actually used, as recommended.*

**One comment about Table 1, which presents over 50 attributes of which the authors only use ~20 in their modelling. Could the table be reduced to present only the used attributes or is the introduction of the other attributes necessary?**

**Answer:** *We appreciate the reviewer's valuable suggestion. Although Table 1 lists over 50 attributes, all 48 morphometric covariates derived from the digital elevation model were initially included as predictor variables in our modeling. The Random Forest algorithm inherently performs variable selection by prioritizing those variables that contribute most to the prediction accuracy. Considering similar feedback from other reviewers requesting the presentation of all input variables for transparency and reproducibility, we have opted to maintain the full list of attributes in Table 1. This approach ensures that readers can fully understand the scope of data considered in our analyses.*

*This aspect is thoroughly addressed in the Methodology section, as detailed below:*

*"The covariate selection based on importance is executed using the backward-forward method, employing the Recursive Feature Elimination (RFE) function available in the "caret" package (Kuhn and Johnson, 2013). This RFE technique is algorithm-specific and yields an optimal set of covariates utilized in predicting the final model for each respective*

*algorithm. RFE is a selection procedure that iteratively removes variables contributing the least to the model, employing an importance measure tailored to each algorithm (Kuhn and Johnson, 2013)."*

---

## Author Response (AR2)

Dear Editor and Reviewers,

Kind regards,

We sincerely appreciate the reviewer's valuable contributions to improving the quality of our manuscript. We have implemented all suggested changes from the minor review (from pdf file), which have substantially enhanced the clarity and overall quality of the manuscript. Additionally, we revised the title to make it more concise and better emphasize the focus of our study. We also improved the keywords and included an introductory concept of pedogeophysics.

We remain at your disposal for any further questions or suggestions for improvement.

---

## Author Response (AR3)

**Dear Editor,**

We sincerely appreciate the opportunity to submit our work to this esteemed journal.

All the requested revisions have been carefully addressed, and the authors' names have been written in full as instructed.

Sincerely,